# Multiobjective Optimization Method of Solid Rocket Motor Finocyl Grain Based on Surrogate Model

**Qiuwen Miao** [1,2], **Zhibin Shen** [1,2,*], **Huihui Zhang** [1,2] and **Haitao Sun** [1,2]

1   College of Aerospace Science and Engineering, National University of Defense Technology, Changsha 410073, China
2   Hunan Key Laboratory of Intelligent Planning and Simulation for Aerospace Missions, Changsha 410073, China
*   Correspondence: zb_shen@yeah.net

**Abstract:** To improve the performance of a solid rocket motor (SRM), a multiobjective optimal design method that can consider the structural integrity, internal ballistic performance, and loading performance of the SRM was proposed based on parametric modeling and surrogate modeling technology. Firstly, the parametric modeling technology was introduced into the field of structural integrity analysis for a high-loading SRM, based on which the influences of load and geometric parameters on the maximum von Mises strain of the SRM grain were analyzed, which effectively improved the sampling speed and prediction accuracy of the surrogate model. Combining the calculation models of the combustion surface area and volume loading fraction of the SRM, the Pareto optimal solution set was obtained based on the NSGA-II algorithm. Under the constraints of the optimization model, the maximum von Mises strain can be reduced by up to 26.72% and the volume loading fraction can be increased by up to 1.83% compared with the original. In addition, the optimal design method proposed in this paper is significantly superior in efficiency, capable of reducing both the single sampling time by more than 95% and the number of numerical simulations from 20,000 to 400, and the average prediction deviation is only 1.87%.

**Keywords:** solid rocket motor; multiobjective optimization; surrogate model; parametric modeling; grain structural integrity

## 1. Introduction

The new generation of missile weapon systems puts forward the performance requirements of high loading, high-pressure strength, and high reliability for SRM. However, high-loading SRMs are always accompanied by high stress and strain, which brings serious challenges to its grain structural integrity. The multiobjective optimization design of the grain shape of the SRM is an effective method to balance the conflicts between the grain structural integrity, the internal ballistic performance, and the loading performance. However, the process of optimizing the SRM charge shape is repetitive, and obtaining the structural response of the SRM through numerical analysis is an extremely time-consuming job. For this reason, there is an urgent need to explore efficient multiobjective optimization design methods to improve the level of design of complex high-loading SRMs.

At present, the parametrized modeling method has been mainly applied to the internal ballistic calculation models and structural analysis models of some simple SRMs [1–3]. Some scholars have carried out a series of works based on CATIA, Pro/E, SolidWorks, and other software for parametrized modeling of SRMs [3–5] and realized the calculation and optimization for the internal ballistic performance of SRMs with various grain shapes [6–8]. Some scholars have also introduced parametrized modeling methods into the field of structural analysis of SRMs [9–11] and applied them to some SRMs with low loading fraction and relatively simple grain shapes.

However, the parametrized modeling method for high-loading SRMs is still lacking in-depth research. The reason for this is that the realization of the analysis process of

high-loading SRMs has a drastic conflict between the complex configuration of the grain and the limited preprocessing capability of the finite element software.

As for the surrogate modeling technique, it has been widely used for various structural optimization design problems since it was proposed in the 1960s [12,13], and some scholars have conducted some preliminary studies for SRM. Ye et al. [14] conducted a multidisciplinary optimization design for SRM based on the Kriging model, considering the component cost and the internal ballistic performance; Wu et al. [15] applied the surrogate modeling technique to the grain design field of SRM and obtained the smoothest charge design solution for the thrust.

As for the multiobjective optimal design of SRMs, Miao et al. [16] carried out a multiobjective optimal design of the stress release boot for several performance indicators related to the structural integrity of an SRM. Tola [17] carried out a multiobjective optimal design of a simple two-dimensional star-hole-type grain considering both internal ballistic performance and structural strength criteria.

In summary, for the optimization design of the grain of a high-loading SRM, the complex structural form causes great difficulties in obtaining the structural response data, and there is still a lack of efficient optimization methods that consider the structural integrity, internal ballistic performance, and loading performance of the SRM grain.

In this paper, the implementation of parametrized modeling methods in the field of structural analysis of a high-loading SRM is investigated based on Abaqus software and Python language. On this basis, the influences of geometric parameters and loads on the structural integrity of the grain are analyzed for the SRM with finocyl grain. Then, the applicability of different experimental design methods and surrogate models to the SRM structure optimization problem was analyzed and tested, and the process of establishing the surrogate model for structure optimization was finalized. Finally, the NSGA-II algorithm was used to realize the multiobjective optimization design of the SRM grain by combining the internal ballistic and loading performance calculation models.

## 2. Model and Methods

In order to implement the multiobjective optimization design of SRM grain, we first need to determine the design variables, objective functions, and constraints, and then establish the mathematical models after transforming it into mathematical problem.

### 2.1. Optimized Objects

The SRM with finocyl grain is widely used in industry for its advantages of high charge–volume fraction and flexible combustion surface adjustment, etc. [18]. We take a fixed-volume SRM with finocyl grain with an aspect ratio of 5.6 and a diameter of 1.7 m as the research object, whose case dimensions have been fixed. The grain is divided into parts I, II, and III, as shown in Figure 1.

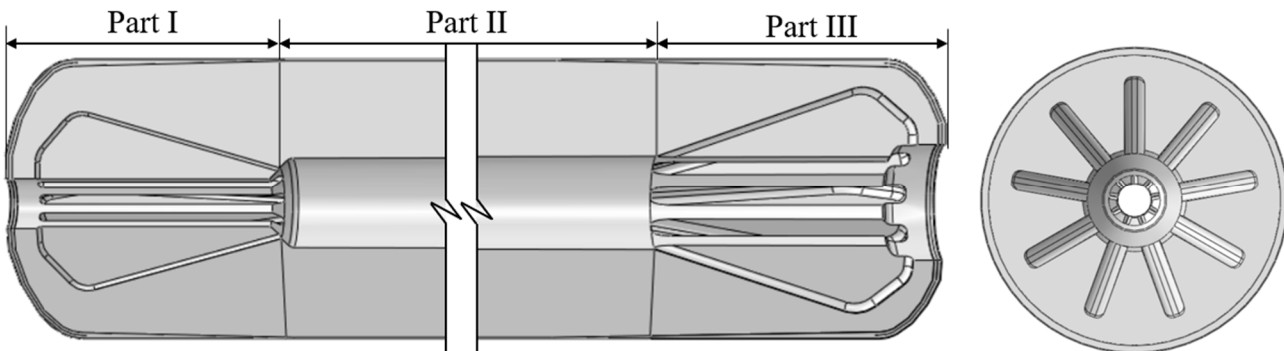

**Figure 1.** Geometric model of the SRM with finocyl grain.

The SRM is in the stage of optimization design of the grain shape after the overall scheme is determined, its internal ballistic performance is basically confirmed, and the case size, grain thickness, and internal hole radius are basically fixed. Therefore, the design variables in this paper are the eighteen configuration parameters associated with finocyl in grain parts I and III, that is,

$$X = (L, R, l_1, l_2, r_1, r_2, \theta, \theta_1, \theta_2, L', R', l'_1, l'_2, r'_1, r'_2, \theta', \theta'_1, \theta'_2) \tag{1}$$

where $L, R, \theta, l_1, l_2, r_1, r_2, \theta_1, \theta_2$ is the parameter of the finocyl in grain part III; the specific meaning of the parameter is shown in Figure 2; $L', R', l'_1, l'_2, r'_1, r'_2, \theta', \theta'_1, \theta'_2$ is the parameter of the finocyl in grain part I; while the initial value $X_0$ and the range $[X_L, X_U]$ of design variables are shown in Table 1.

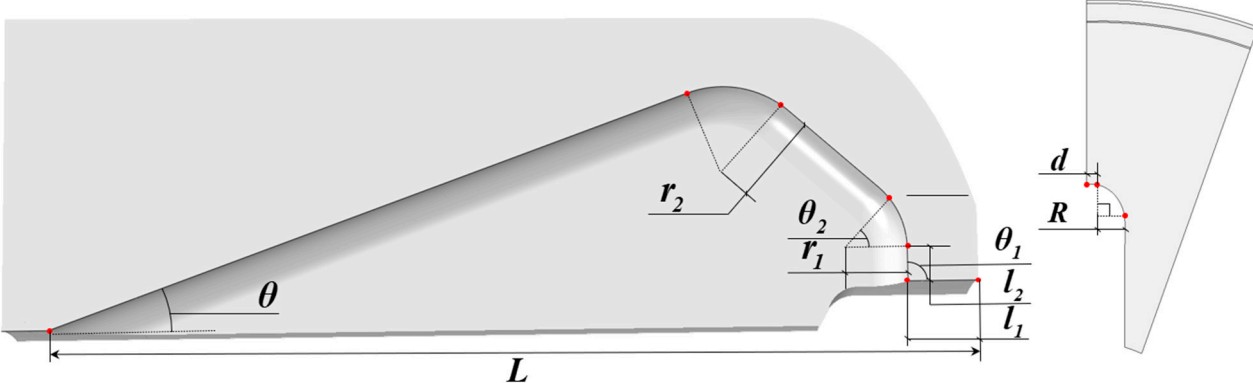

**Figure 2.** Schematic diagram of the $L, R, \theta, l_1, l_2, r_1, r_2, \theta_1, \theta_2$ in grain part III.

**Table 1.** Initial values and range of values for the design variables of the SRM grain.

| Variable | $L$/mm | $R$/mm | $l_1$/mm | $l_2$/mm | $r_1$/mm | $r_2$/mm | $\theta$/° | $\theta_1$/° | $\theta_2$/° |
|---|---|---|---|---|---|---|---|---|---|
| $x_L$ | 500.0 | 3.0 | 30.0 | 5.0 | 40.0 | 60.0 | 10.0 | 80.0 | 40.0 |
| $x_0$ | 739.2 | 18.0 | 54.3 | 25.0 | 60.0 | 80.0 | 19.0 | 90.0 | 49.0 |
| $x_U$ | 900.0 | 27.0 | 60.0 | 60.0 | 80.0 | 100.0 | 30.0 | 90.0 | 90.0 |
| Variable | $L'$/mm | $R'$/mm | $l'_1$/mm | $l'_2$/mm | $r'_1$/mm | $r'_2$/mm | $\theta'$/° | $\theta'_1$/° | $\theta'_2$/° |
| $x_L$ | 350.0 | 4.0 | 5.0 | 5.0 | 40.0 | 40.0 | 10.0 | 80.0 | 40.0 |
| $x_0$ | 688.0 | 10.0 | 42.5 | 25.0 | 60.0 | 60.0 | 16.1 | 90.0 | 40.7 |
| $x_U$ | 800.0 | 16.0 | 60.0 | 60.0 | 80.0 | 80.0 | 30.0 | 90.0 | 90.0 |

### 2.2. Optimization Objectives and Models

For the SRM, structural integrity, internal ballistic performance, and loading performance are the three key performances for determining its launch capability.

1. Structural integrity

The main failure mode of SRM is the destruction of its structural integrity, especially the structural integrity of its grain. From the beginning of production to the completion of the launch mission, its life history and the corresponding loads are shown in Figure 3. Long-term experience indicates that temperature and pressure loads have the most significant impact on the structural integrity of the SRM, so the load case of the SRM studied in this paper is a combined load consisting of temperature and pressure.

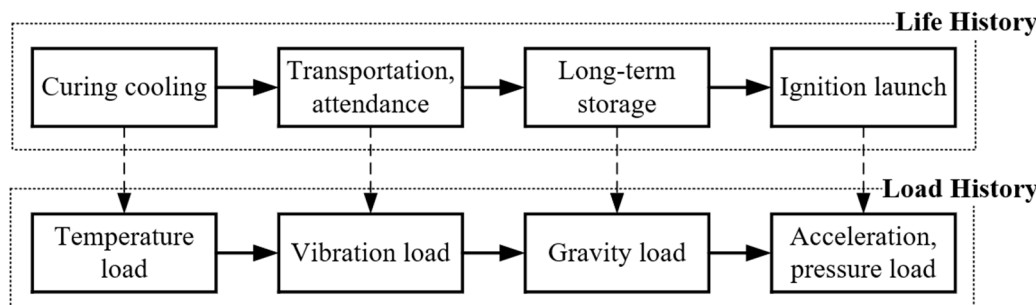

**Figure 3.** Schematic diagram of the life and load history of SRM.

The SRM grain is in a multidirectional stress state under temperature or pressure load, so it is appropriate to use the octahedral shear strain criterion, which is $r_8 \leq \frac{r_{8m}}{n}$, where $r_{8m}$ is the critical value obtained by the experiment and $n$ is the safety factor.

The expression for the octahedral shear strain is given by

$$r_8 = \frac{1}{3}\sqrt{(\varepsilon_x - \varepsilon_y)^2 + (\varepsilon_y - \varepsilon_z)^2 + (\varepsilon_z - \varepsilon_x)^2 + 6(\varepsilon_{xy}^2 + \varepsilon_{yz}^2 + \varepsilon_{zx}^2)} \tag{2}$$

and the expression of von Mises strain is

$$\varepsilon_v = \frac{\sqrt{2}}{3}\sqrt{(\varepsilon_x - \varepsilon_y)^2 + (\varepsilon_y - \varepsilon_z)^2 + (\varepsilon_z - \varepsilon_x)^2 + 6(\varepsilon_{xy}^2 + \varepsilon_{yz}^2 + \varepsilon_{zx}^2)} \tag{3}$$

Therefore, $\varepsilon_v = \frac{\sqrt{2}}{2}r_8$. It can be obtained that the von Mises strain is essentially equivalent to the octahedral shear strain. In addition, for solid propellants with Poisson ratio tending to 0.5, the results of the monotonic tensile test can be applied more simply when using the von Mises strain criterion.

For the SRM, the lower the von Mises strain, the safer the structure of SRM. For this reason, the objective function related to the structural integrity of the SRM was set as

$$\min f_{\varepsilon_{v,\max}}(X) = \frac{[\varepsilon_{v,\max}(X) - \varepsilon_{v,\max}^0]}{\varepsilon_{v,\max}^0} \times 100\% \tag{4}$$

where $f_{\varepsilon_{v,\max}}(X)$ is a function of the maximum von Mises strain $\varepsilon_{v,\max}$ about $X$, $\varepsilon_{v,\max}(X)$ is the $\varepsilon_{v,\max}$ for different $X$, and $\varepsilon_{v,\max}^0$ is the maximum von Mises strain before optimization.

2. Internal ballistic performance

For the internal ballistic performance of a SRM, the combustion chamber pressure is one of the most critical parameters, because its magnitude and variation law not only directly determine the SRM thrust scheme, but also affect the whole process of combustion and the design of the structural strength of the SRM.

In zero-dimensional internal ballistics, assuming a uniform pressure in the combustion chamber and ignoring the effect of gas flow [19], the basic differential equation for the time variation of the combustion chamber pressure can be obtained according to the conservation of mass and the gas equation of state as

$$\frac{V_c}{\Gamma^2 c^{*2}} \frac{dp_c}{dt} = p_p A_b \alpha p_c^n - \frac{p_c A_t}{c^*} \tag{5}$$

where $V_c$ is the free volume of the combustion chamber, $\Gamma$ is the specific heat ratio function, $c^*$ is the characteristic velocity of the propellant, $p_c$ is the gas density, $t$ is the time, $p_p$ is the propellant density, $A_b$ is the area of the combustion surface, $\alpha, n$ is the calculated correlation coefficient of the combustion velocity, and $A_t$ is the cross-sectional area of the nozzle throat.

For the SRM that has completed the preliminary design of internal ballistics, the combustion chamber shape, propellant type, and nozzle design have been determined, so the change in combustion chamber pressure and combustion surface area with time is

basically the same. For this reason, the objective function related to the internal ballistic performance is set as follows: the deviation of the initial combustion surface area from the target combustion surface area is minimized, that is,

$$\min f_{A_b}(X) = \frac{\left| A_b(X) - A_b^0 \right|}{A_b^0} \times 100\% \tag{6}$$

where $f_{A_b}(X)$ is a function of the initial combustion surface area $A_b$ about $X$, $A_b(X)$ is the initial combustion surface area for different $X$, and $A_b^0$ is the initial combustion surface area before optimization.

3. Loading performance

The volume loading fraction reflects the loading performance of the SRM when the combustion chamber size is constant, and the increase in propellant loading can directly improve the total impulse, so the volume loading fraction should be increased as much as possible during the design process. Therefore, the objective function related to the loading performance is set as follows: maximize the increase in the volume loading fraction relative to that before optimization, that is,

$$\max f_{\eta_v}(X) = \frac{\left[ \eta_v(X) - \eta_v^0 \right]}{\eta_v^0} \times 100\% \tag{7}$$

where $f_{\eta_v}(X)$ is a function of the volume loading fraction $\eta_v$ about $X$, $\eta_v(X)$ is the volume loading fraction for different $X$, and $\eta_v^0$ is the volume loading fraction before optimization.

In addition, the optimization objectives should also satisfy certain conditions, namely that the maximum von Mises strain of the SRM grain should not be higher than before the optimization, the volume filling fraction of the SRM should not be lower than before the optimization, and the deviation between the initial combustion surface area and the target combustion surface area should be within 5%.

In summary, the mathematical model of the multiobjective optimization problem of the SRM grain is

$$\begin{cases} \min f_{\varepsilon_{v,\max}}(X) \\ \min f_{A_b}(X) \\ \max f_{\eta_v}(X) \\ s.t. \ f_{\varepsilon_{v,\max}}(X) \le 0.00\% \\ \quad f_{A_b}(X) \le 5.00\% \\ \quad f_{\eta_v}(X) \ge 0.00\% \\ \quad X \in [X_L, X_U], X = (L, R, \theta, l_1, l_2, r_1, r_2, \theta_1, \theta_2, L', R', \theta', l'_1, l'_2, r_1', r'_2, \theta'_1, \theta'_2) \end{cases} \tag{8}$$

## 3. Grain Structural Integrity Analysis Based on Parametric Modeling Technology

The key and difficulty of the multiobjective optimization of the grain shape is how to obtain the structural response under different design variables. The optimization process requires repeated invocations of the numerical simulation model to obtain the structural response data, and the high-precision simulation analysis model is always accompanied by high computational costs, resulting in long optimization cycles and low solution efficiency.

In order to improve the optimization efficiency, the parametric modeling method was first introduced into the field of structural analysis of a high-loading SRM, and it was further combined with the surrogate modeling technique to establish the mapping relationship between input and output data through limited sample data, which can effectively reduce the total computational effort of the optimization process.

### 3.1. Parametric Modeling Technology for SRM

The structural integrity analysis of the grain of the SRM is a really complex and time-consuming process, and the manual modeling time is about tens of hours. For this reason, a

Python script is used to replace the user's pre- and postprocessing operations, which reduces the cost of the structural analysis to tens of minutes; the basic principle is shown in Figure 4.

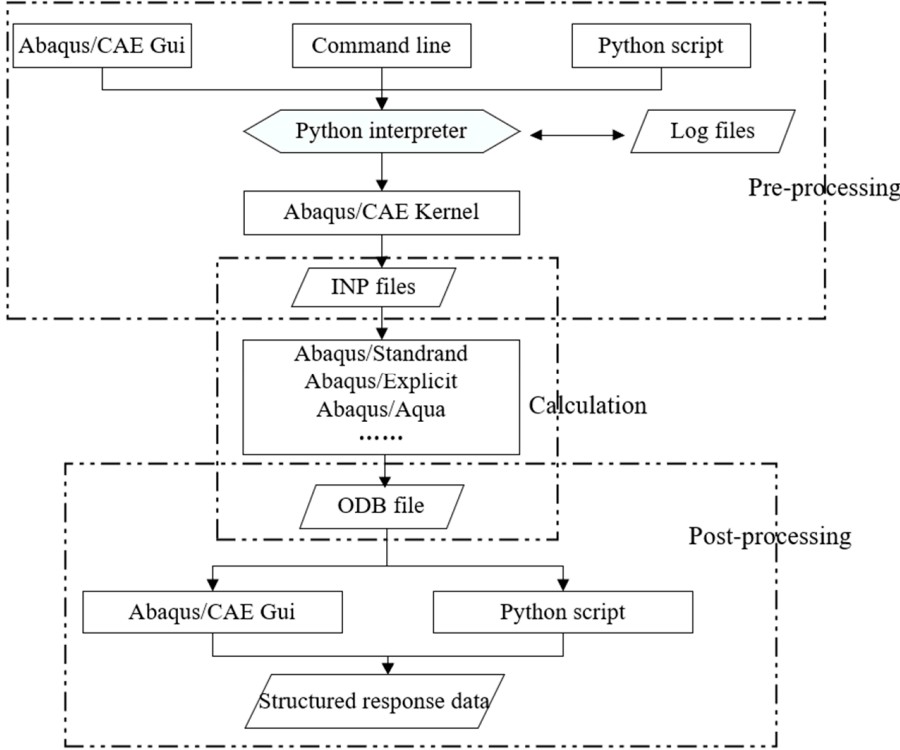

**Figure 4.** Schematic diagram of parametric modeling technology based on Abaqus and Python.

The Python script for the SRM mainly contains six functions: geometric modeling, meshing, material definition, boundary setting, load setting, and result extraction. The content and order of the script are shown in Figure 5.

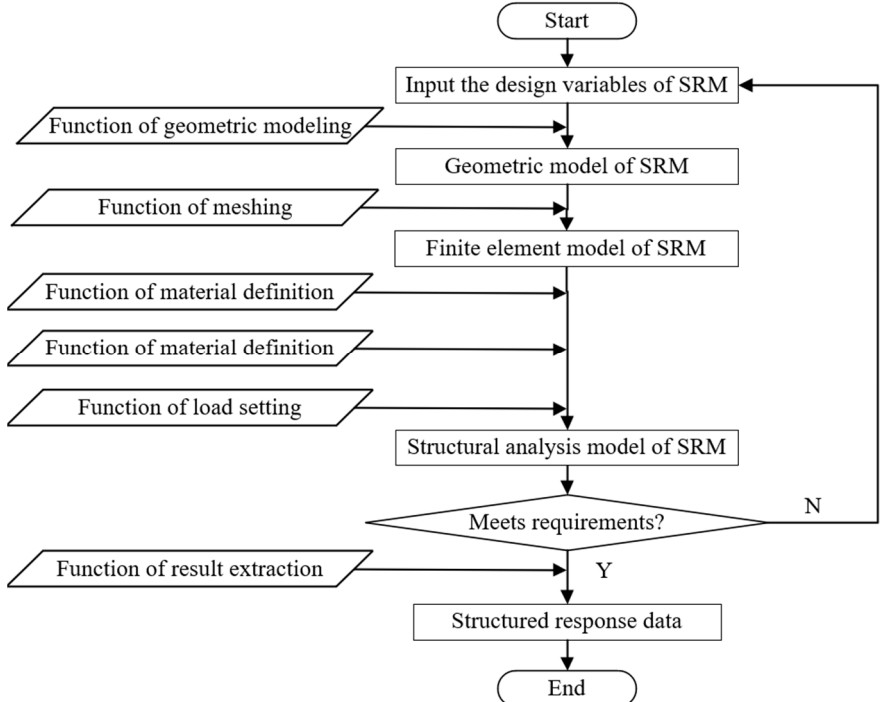

**Figure 5.** Flowchart for parametric modeling of SRM based on Python functions.

The specific features of the six functions are:

1. Function of geometric modeling: since the SRM grain has nine holes with the same circumferential position on the front and rear sides, combined with its cyclic symmetry, the 1/18 geometric model, as shown in Figure 6a, will be created by the function of geometric modeling.

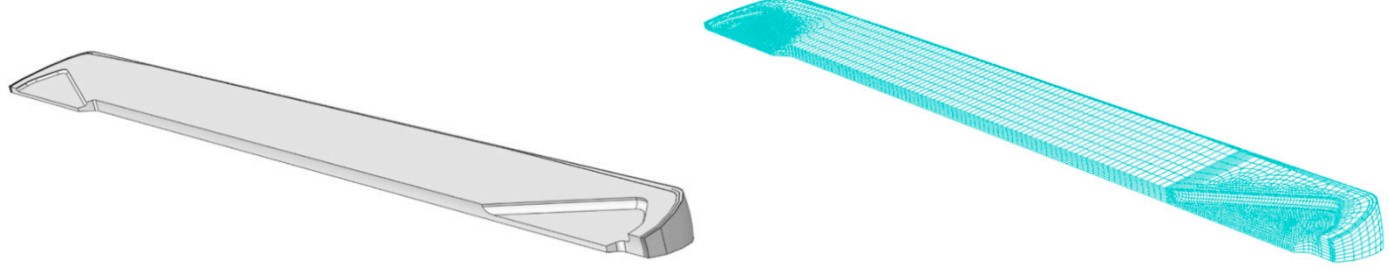

(**a**) geometry model      (**b**) Hexahedral mesh finite element model

**Figure 6.** The 1/18th model of the SRM with finocyl grain.

2. Function of meshing: dividing the geometric model automatically into the finite element model shown in Figure 6b, which has the number of hexahedral elements at around 150,000.

3. Function of material definition: assigning the material properties of each part to the corresponding geometry. The material parameters of the case, insulation, cladding, and propellant are shown in Table 2. The propellant and cladding are the same viscoelastic materials, and their relaxation modulus is characterized using the Prony series as shown in Equation (9), and the first five order parameters are shown in Table 3.

$$E(t) = E_0 - \sum_{i=1}^{n} E_i \left( 1 - e^{-t/\tau_i} \right) \tag{9}$$

**Table 2.** Material parameters of the SRM.

| Material Parameters | Elastic Modulus/MPa | Poisson's Ratio | Density/(kg/m³) | Expansion Coefficient/$^\circ$C$^{-1}$ |
|---|---|---|---|---|
| Case | $1.86 \times 10^5$ | 0.300 | $7.9 \times 10^3$ | $1.10 \times 10^{-5}$ |
| Insulator | 60.00 | 0.498 | $2.1 \times 10^3$ | $2.20 \times 10^{-5}$ |
| Cladding | Prony series | 0.498 | $1.86 \times 10^3$ | $8.60 \times 10^{-5}$ |
| Propellant | Prony series | 0.498 | $1.86 \times 10^3$ | $8.60 \times 10^{-5}$ |

**Table 3.** The first 5 order parameters of Prony series.

| $i$ | 0 | 1 | 2 | 3 | 4 | 5 |
|---|---|---|---|---|---|---|
| $\tau_i$ | | 0.0037 | 0.1099 | 3.2977 | 98.9295 | 2967.8852 |
| $E_i$ | 18.2082 | 8.3728 | 4.2322 | 2.3112 | 0.9354 | 0.8212 |

The time–temperature equivalence equation of propellant and cladding is shown in Equation (10), where $\alpha_T$ is the translational factors at different temperatures; $C_1$, $C_2$ is the material constant, determined by the material's own characteristics, $C_1 = 5.0474$, $C_2 = 144.9207$; $T_0$ is the reference temperature, $T_0 = 293.15$.

$$\lg \alpha_T = \frac{-C_1(T - T_0)}{C_2 + (T - T_0)} \tag{10}$$

4. Function of boundary setting: applying symmetric displacement constraints on both sides of the SRM according to the cyclic symmetry, and then applying axial displacement constraints on the intersection of the tail head and the straight section of the case.

5. Function of load setting: setting the analysis step and loading the load to the finite element model of the SRM. When the temperature load is applied, a linear decrease in temperature from 58 °C to 20 °C in 86,400 s is placed on the SRM; when the internal pressure load is applied, a linear increase in pressure to 10 MPa in 0.3 s is placed on the internal surface of the SRM [20].

### 3.2. Analysis of Influence of Load

Elasticity theory indicates that for the plane strain model, the stress concentration coefficients caused by uniform temperature load and uniform internal pressure load are the same. For the linear viscoelastic three-dimensional grain model used in this paper, the grain will shrink under the temperature load, which was similar to the surrounding pressure environment in which the pressure load was applied, and the von Mises stress cloud of the grain when the temperature and pressure load are applied separately is shown in Figure 7.

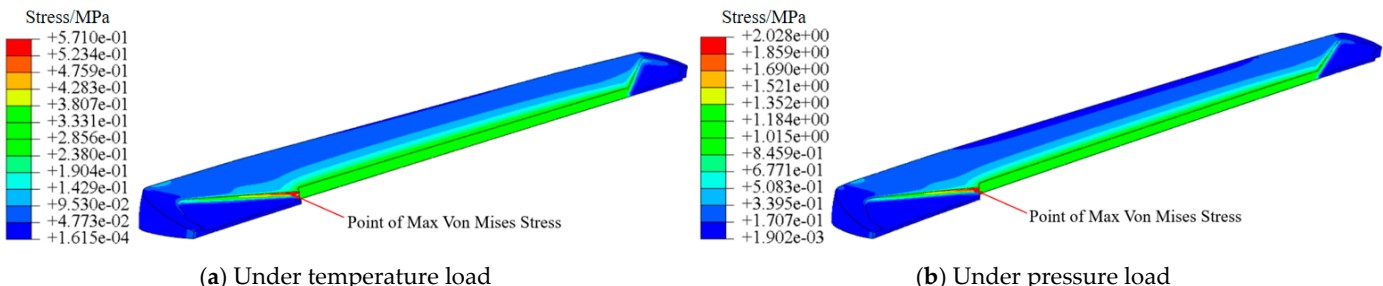

(**a**) Under temperature load　　　　　　　　　　　　　(**b**) Under pressure load

**Figure 7.** Von Mises stress cloud of the grain under different load.

From Figure 7, it can be obtained that the stress cloud diagrams of the grain are extremely similar when the temperature load and the pressure load are applied separately, and the locations of the danger points of each part of the grain are exactly the same, in which the danger points of grain part I and III appear near the junction point between the tail of the finocyl and the inner hole, while the danger points of grain part I appear on the central surface of the inner hole, and the maximum von Mises strains of three parts are described as $\varepsilon_{v,\max}^{\mathrm{I}}, \varepsilon_{v,\max}^{\mathrm{II}}, \varepsilon_{v,\max}^{\mathrm{III}}$, respectively.

The $\varepsilon_{v,\max}^{\mathrm{III}}$ was taken as the object of study, keeping the rest parameter fixed and making $L$ and $\theta$ change in their respective ranges, while the $\varepsilon_{v,\max}^{\mathrm{III},t}$ under temperature load, the $\varepsilon_{v,\max}^{\mathrm{III},p}$ under pressure load and the $\varepsilon_{v,\max}^{\mathrm{III},t+p}$ under combined load were extracted, respectively, as shown in Figure 8.

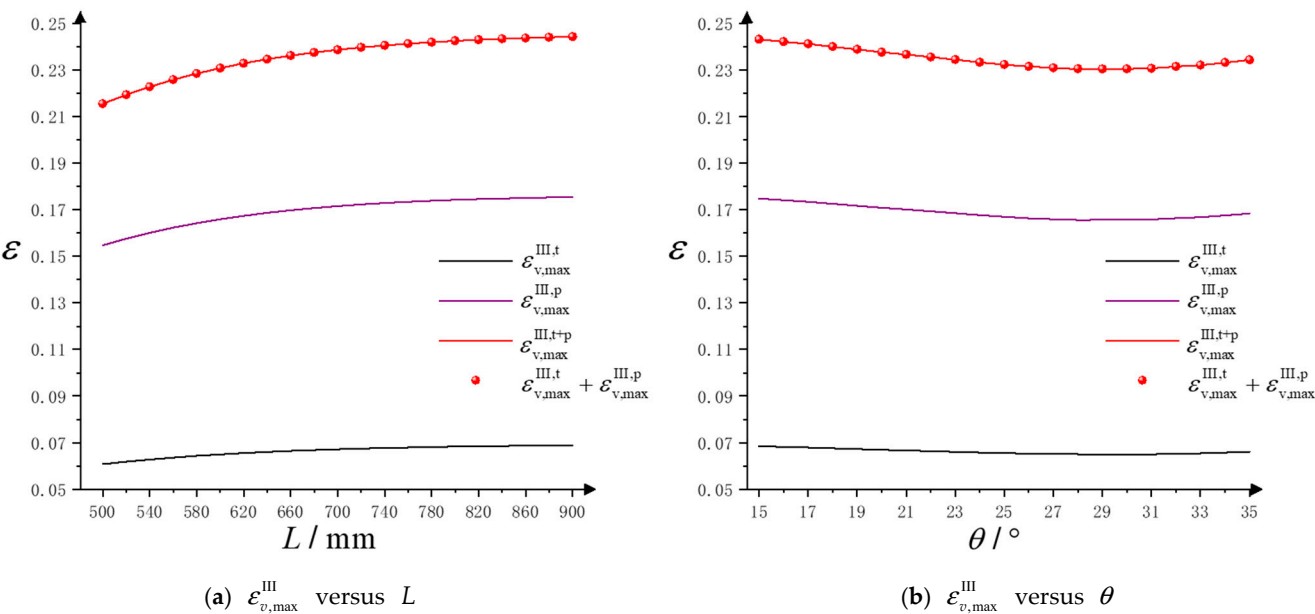

**Figure 8.** Schematic diagram of the superposition analysis of load influence.

From Figure 8, the effects of temperature and pressure load are almost directly superimposable for $\varepsilon_{v,\max}^{\mathrm{III},t} + \varepsilon_{v,\max}^{\mathrm{III},p} \approx \varepsilon_{v,\max}^{\mathrm{III},t+p}$. From von Mises strain Equation (3), it is obtained that

$$\varepsilon_{v,\max}^{\mathrm{III},t+p} = \frac{\sqrt{2}}{3}\sqrt{\left(\varepsilon_1^{\mathrm{III},t+p} - \varepsilon_2^{\mathrm{III},t+p}\right)^2 + \left(\varepsilon_1^{\mathrm{III},t+p} - \varepsilon_3^{\mathrm{III},t+p}\right)^2 + \left(\varepsilon_2^{\mathrm{III},t+p} - \varepsilon_3^{\mathrm{III},t+p}\right)^2} \tag{11}$$

Combined with the principal strain situation of the grain, for linear viscoelastic materials, the position of $\varepsilon_{v,\max}^{\mathrm{III}}$ and the direction of the principal strain under the effect of temperature or pressure load are completely same. For this reason, the principal strain under combined load is equal to the sum of the principal strains when temperature and pressure loads are applied separately, that is,

$$\varepsilon_i^{\mathrm{III},t+p} = \varepsilon_i^{\mathrm{III},t} + \varepsilon_i^{\mathrm{III},p} \, (i = 1, 2, 3) \tag{12}$$

where $\varepsilon_i^{\mathrm{III},t}, \varepsilon_i^{\mathrm{III},p}, \varepsilon_i^{\mathrm{III},t+p}$ are the principal strains at the danger point under the effect of temperature or pressure load alone or in combination, respectively. Obviously, when the principal strains satisfy

$$\varepsilon_i^{\mathrm{III},t} / \varepsilon_i^{\mathrm{III},p} = K \tag{13}$$

then

$$\varepsilon_{v,\max}^{\mathrm{III},t} = \frac{\sqrt{2}}{3}K\sqrt{\left(\varepsilon_1^{\mathrm{III},p} - \varepsilon_2^{\mathrm{III},p}\right)^2 + \left(\varepsilon_1^{\mathrm{III},p} - \varepsilon_3^{\mathrm{III},p}\right)^2 + \left(\varepsilon_2^{\mathrm{III},p} - \varepsilon_3^{\mathrm{III},p}\right)^2} = K\varepsilon_{v,\max}^{\mathrm{III},p} \tag{14}$$

Therefore,

$$\varepsilon_{v,\max}^{\mathrm{III},t} + \varepsilon_{v,\max}^{\mathrm{III},p} = \frac{\sqrt{2}}{3}(1+K)\sqrt{\left(\varepsilon_1^{\mathrm{III},p} - \varepsilon_2^{\mathrm{III},p}\right)^2 + \left(\varepsilon_1^{\mathrm{III},p} - \varepsilon_3^{\mathrm{III},p}\right)^2 + \left(\varepsilon_2^{\mathrm{III},p} - \varepsilon_3^{\mathrm{III},p}\right)^2} = (1+K)\varepsilon_{v,\max}^{\mathrm{III},p} \tag{15}$$

and from Equations (11) and (12),

$$\varepsilon_{v,\max}^{\mathrm{III},t+p} = \frac{\sqrt{2}}{3}(1+K)\sqrt{\left(\varepsilon_1^{\mathrm{III},p} - \varepsilon_2^{\mathrm{III},p}\right)^2 + \left(\varepsilon_1^{\mathrm{III},p} - \varepsilon_3^{\mathrm{III},p}\right)^2 + \left(\varepsilon_2^{\mathrm{III},p} - \varepsilon_3^{\mathrm{III},p}\right)^2} = (1+K)\varepsilon_{v,\max}^{\mathrm{III},p} \tag{16}$$

Combining Equations (15) and (16), we can obtain

$$\varepsilon_{v,\max}^{\mathrm{III},t} + \varepsilon_{v,\max}^{\mathrm{III},p} = \varepsilon_{v,\max}^{\mathrm{III},t+p} \tag{17}$$

The numerical analysis result shows that under the material parameters and load conditions of this paper, the ratio of principal strain under temperature and pressure load is approximately equal to 0.39, that is,

$$\varepsilon_i^{\mathrm{III},t} / \varepsilon_i^{\mathrm{III},p} \approx K = 0.39 (i = 1,2,3) \tag{18}$$

Therefore,

$$\varepsilon_{v,\max}^{\mathrm{III},t} + \varepsilon_{v,\max}^{\mathrm{III},p} \approx \varepsilon_{v,\max}^{\mathrm{III},t+p} \approx (1+K)\varepsilon_{v,\max}^{\mathrm{III},p} \tag{19}$$

For this reason, when it is required to obtain $\varepsilon_{v,\max}^{\mathrm{III},t+p}$ only the numerical simulation of temperature load or pressure load can be performed, which can save half of the time for the obtaining of the structural response data.

### 3.3. Analysis of the Influence of Parameters

The Saint-Venant principle indicates that when the magnitude of the force is constant, the distribution of the load has little effect on the area far from the effect of the load. For this reason, grain part III shown in Figure 2 was studied, and the parameters were grouped into those close to the danger point—$L, d, R, \theta$—and those far from the danger point—$l_1, l_2, r_1, r_2, \theta_1, \theta_2$.

Firstly, $l_1, l_2, r_1, r_2, \theta_1, \theta_2$ were analyzed, keeping $L, d, R, \theta$ constant, and adjusting $l_1, l_2, r_1, r_2$ and $\theta_1, \theta_2$, respectively, the variation of $\varepsilon_{v,\max}^{\mathrm{I},p}, \varepsilon_{v,\max}^{\mathrm{II},p}, \varepsilon_{v,\max}^{\mathrm{III},p}$ with $l_1, r_1, r_2$ and $\theta_1, \theta_2$ are shown in Figures 9 and 10, in which the left and right cloud maps are the configuration and von Mises stress cloud maps of the grain with the first and last group of parameters, respectively.

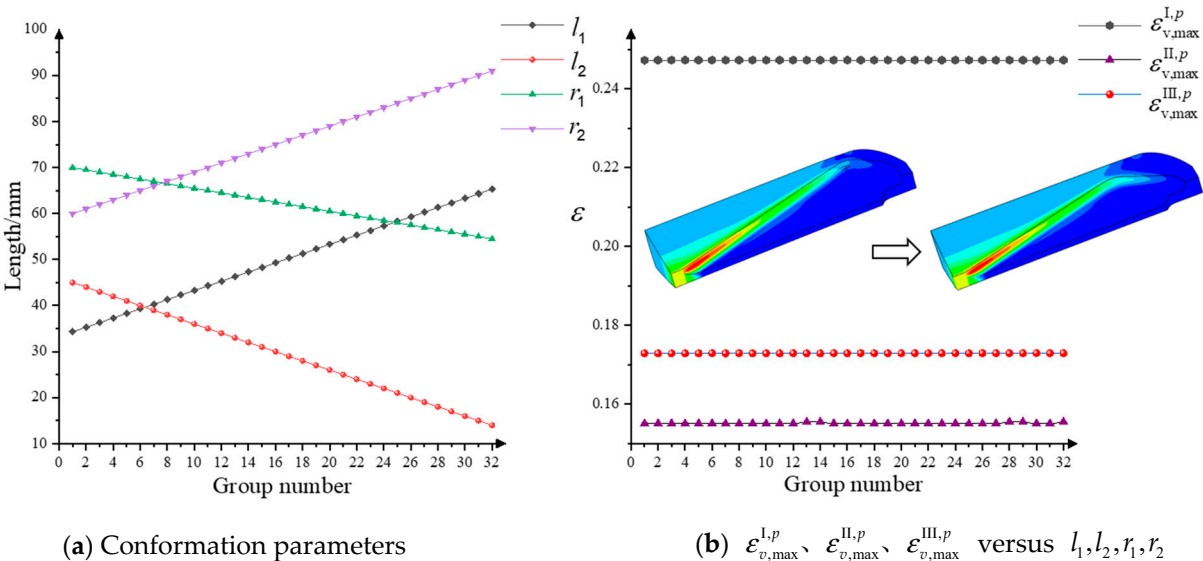

(**a**) Conformation parameters

(**b**) $\varepsilon_{v,\max}^{\mathrm{I},p}$、$\varepsilon_{v,\max}^{\mathrm{II},p}$、$\varepsilon_{v,\max}^{\mathrm{III},p}$ versus $l_1, l_2, r_1, r_2$

**Figure 9.** The relationship map between $\varepsilon_{v,\max}^{\mathrm{I},p}, \varepsilon_{v,\max}^{\mathrm{II},p}, \varepsilon_{v,\max}^{\mathrm{III},p}$ and $l_1, l_2, r_1, r_2$.

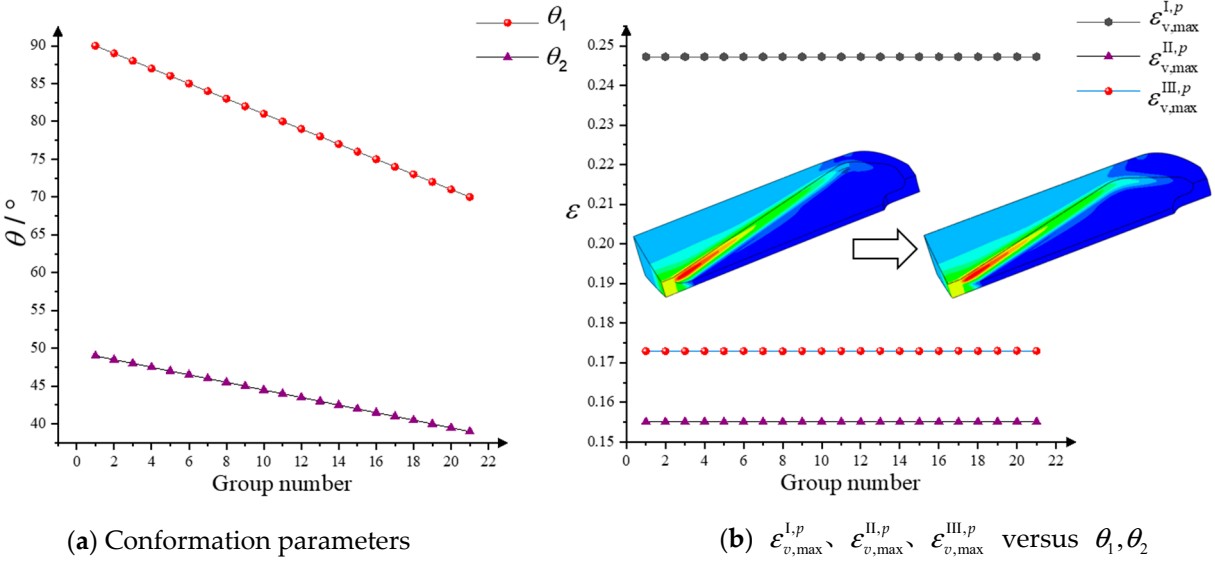

(**a**) Conformation parameters

(**b**) $\varepsilon_{v,\max}^{\mathrm{I},p}$、 $\varepsilon_{v,\max}^{\mathrm{II},p}$、 $\varepsilon_{v,\max}^{\mathrm{III},p}$ versus $\theta_1, \theta_2$

**Figure 10.** The relationship map between $\varepsilon_{v,\max}^{\mathrm{I},p}, \varepsilon_{v,\max}^{\mathrm{II},p}, \varepsilon_{v,\max}^{\mathrm{III},p}$ and $\theta_1, \theta_2$.

From Figures 9 and 10, it can be obtained that $\varepsilon_{v,\max}^{\mathrm{I},p}, \varepsilon_{v,\max}^{\mathrm{II},p}, \varepsilon_{v,\max}^{\mathrm{III},p}$ fluctuate within a very small range when $l_1, l_2, r_1, r_2, \theta_1, \theta_2$ are varied with constant $L, d, R, \theta$, and the positions of the danger points remain basically unchanged. On the one hand, it indicates that the change in $l_1, l_2, r_1, r_2, \theta_1, \theta_2$ does not affect the structural response of grain part I and II, and on the other hand, it also indicates that the change in $l_1, l_2, r_1, r_2, \theta_1, \theta_2$ does not affect the von Mises strain response at the danger point of grain part III.

To analyze the influences of $L, d, R, \theta$, $\varepsilon_{v,\max}^{\mathrm{I}}, \varepsilon_{v,\max}^{\mathrm{II}}, \varepsilon_{v,\max}^{\mathrm{III}}$ at different $L, d, R, \theta$ are shown in Figure 11.

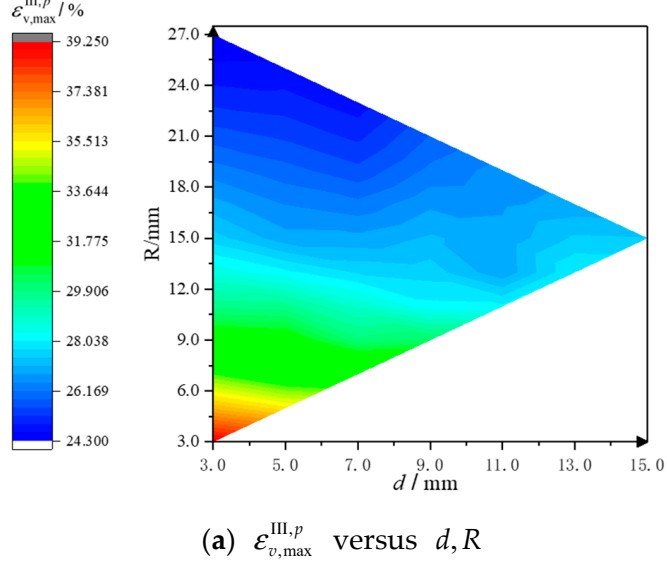

(**a**) $\varepsilon_{v,\max}^{\mathrm{III},p}$ versus $d, R$

**Figure 11.** *Cont.*

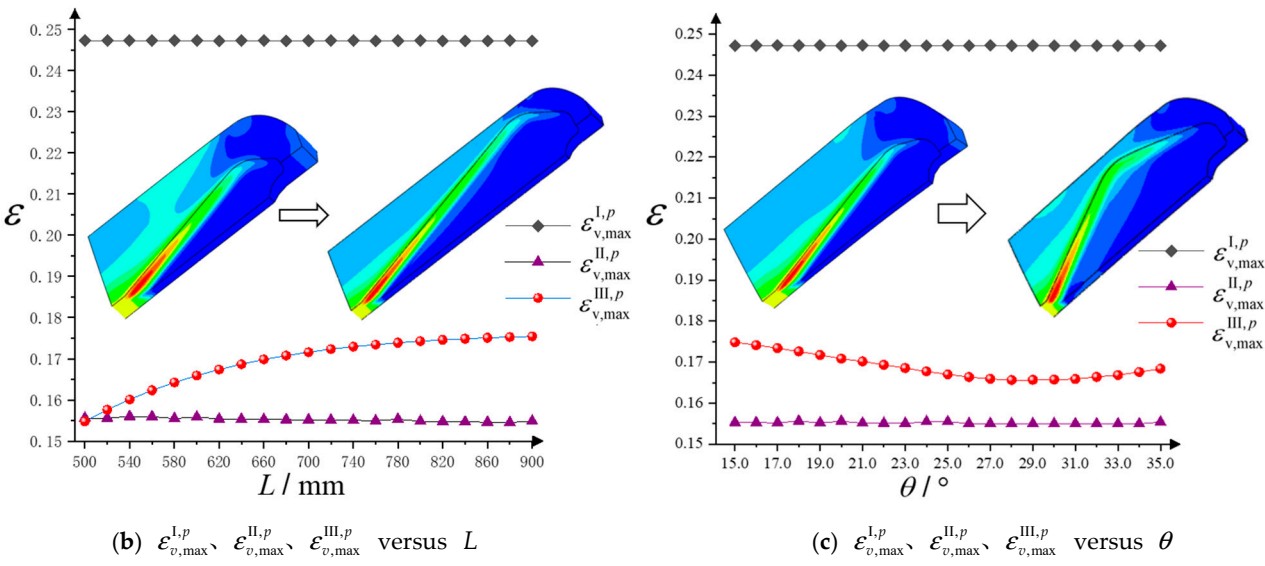

**(b)** $\varepsilon_{v,\max}^{I,p}$、$\varepsilon_{v,\max}^{II,p}$、$\varepsilon_{v,\max}^{III,p}$ versus $L$      **(c)** $\varepsilon_{v,\max}^{I,p}$、$\varepsilon_{v,\max}^{II,p}$、$\varepsilon_{v,\max}^{III,p}$ versus $\theta$

**Figure 11.** The relationship map between $\varepsilon_{v,\max}^{I,p}, \varepsilon_{v,\max}^{II,p}, \varepsilon_{v,\max}^{III,p}$ and $L, d, R, \theta$.

From Figure 11, it can be obtained that:

1. $\varepsilon_{v,\max}^{III,p}$ shows a monotonic decreasing trend with the increase in $d, R$, and the relative increase in $R$ can make $\varepsilon_{v,\max}^{III,p}$ decrease when the sum of $d$ and $R$ is constant;

2. $\varepsilon_{v,\max}^{III,p}$ gradually decreases with the increase in $L$, and the decreasing trend gradually becomes slower;

3. $\varepsilon_{v,\max}^{III,p}$ shows the trend of decreasing and then increasing with the increase in $\theta$, and reaches the minimum value near $\theta = 30°$, and $\varepsilon_{v,\max}^{III,p}$ does not change with $\theta$.

4. $\varepsilon_{v,\max}^{I,p}, \varepsilon_{v,\max}^{II,p}$ also does not change with $L, d, R, \theta$.

In summary, based on the analysis of the effects of loads and parameters, for the acquisition of $\varepsilon_{v,\max}^{III,t+p}$ with different parameters, based on the superposition analysis of load effects in Section 3.2, we can firstly obtain that

$$\varepsilon_{v,\max}^{III,t+p} = (1+K)\varepsilon_{v,\max}^{III,p} \tag{20}$$

After partitioning the SRM as shown in Figure 1, Equation (20) is transformed into

$$\varepsilon_{v,\max}^{t+p} = (1+K)\max(\varepsilon_{v,\max}^{I,p}, \varepsilon_{v,\max}^{II,p}, \varepsilon_{v,\max}^{III,p}) \tag{21}$$

Combined with the conclusions obtained from the parameter influence analysis in Section 3.3, we can obtain

$$\begin{cases} \varepsilon_{v,\max}^{I,p} = f_I^p(d', R', L', \theta') \\ \varepsilon_{v,\max}^{II} = \varepsilon_0^p \\ \varepsilon_{v,\max}^{II,p} = f_{III}^p(d, R, L, \theta) \end{cases} \tag{22}$$

where $\varepsilon_0^p$ is the maximum von Mises strain of part II of grain and $\varepsilon_0^p$ is a constant.

Then, according to conclusion (2) of Section 3.3, controlling $d' = 2.0$ mm and $d = 3.0$ mm in the subsequent analysis, Equation (22) is simplified to

$$\begin{cases} \varepsilon_{v,\max}^{I,t+p} = (1+K)f_I^p(R', L', \theta') \\ \varepsilon_{v,\max}^{II} = (1+K)\varepsilon_0^p \\ \varepsilon_{v,\max}^{II,t+p} = (1+K)f_{III}^p(R, L, \theta) \end{cases} \tag{23}$$

## 4. Establishment and Validation of Strain Surrogate Model

The multiobjective optimization design of SRM grain requires repeated invocation of numerical simulation models to obtain structural response data. High-precision simulation analysis models are always accompanied by high computing costs, resulting in long optimization cycles and low solution efficiency. For this reason, the surrogate model technology

is introduced to establish the mapping relationship between $\varepsilon_{v,\max}^{\mathrm{III},t+p}$ and $X$ to improve the optimization efficiency and reduce the optimization cost.

Based on the conclusion of Section 3.2 and Equations (21) and (23), the establishment and validation process of the surrogate model for $\varepsilon_{v,\max}^{\mathrm{III},t+p}$ is shown in Figure 12.

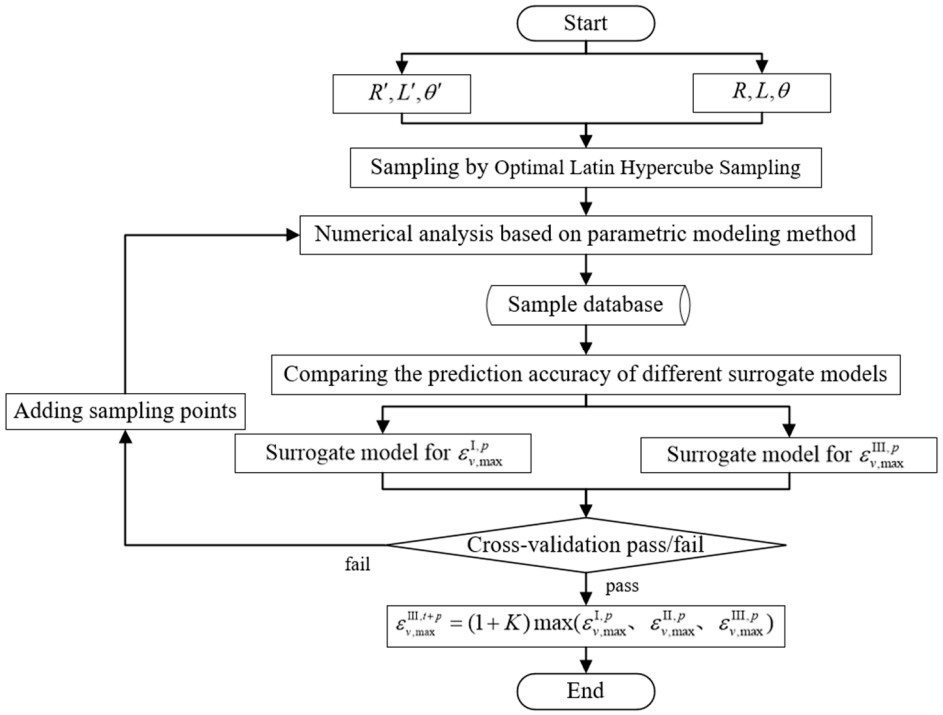

**Figure 12.** Flowchart on the establishment and validation of the surrogate model of $\varepsilon_{v,\max}^{\mathrm{III},t+p}$.

### 4.1. Acquisition of Sample Data

In order to enable the surrogate model to effectively predict the response in the design space, a certain number of initial sample points need to be obtained first. The acquisition of sample points is based on the experimental design method, and to make the surrogate model effectively replace the true objective function, a suitable experimental design method should be selected so that the initial sample points can be distributed as uniformly as possible in the design space.

The most widely used experimental design method is optimal Latin hypercube sampling (OPLHS). OPLHS is based on Latin hypercube sampling (LHS), in which the sampling process will evaluate and optimize different combinations based on the total spacing between sample points using the optimization algorithm, and finally derive the most uniformly distributed combination of sample points.

For this purpose, 200 data samples were collected in the sampling space shown in Figure 1 for grain parts I and III, respectively, and the spatial distribution of the sampling points is shown in Figure 13.

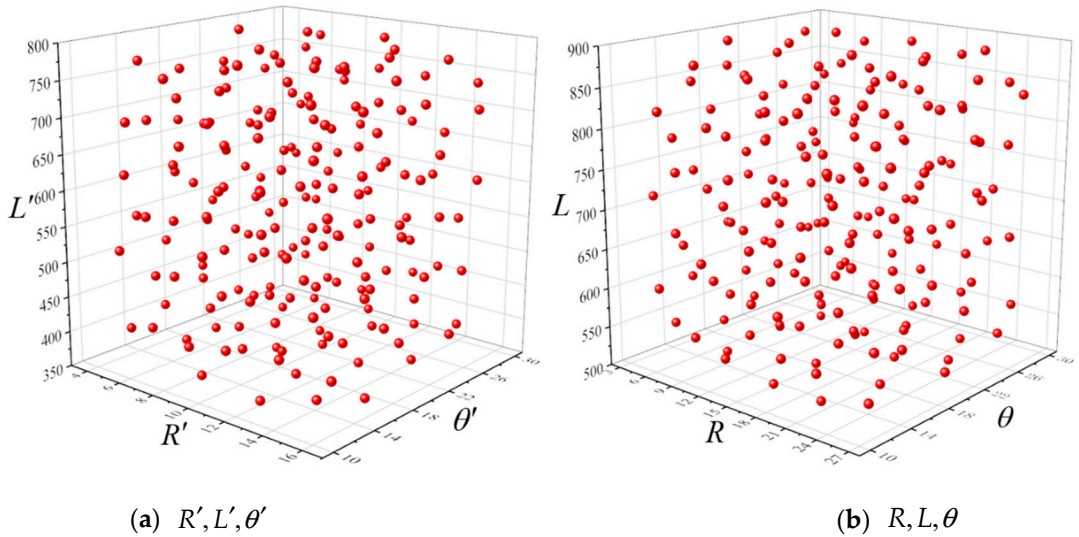

**(a)** $R', L', \theta'$                          **(b)** $R, L, \theta$

**Figure 13.** Schematic diagram of the sampling points based on OPLHS.

*4.2. Comparison and Validation of Surrogate Model*

The commonly used surrogate modeling techniques include the response surface methodology (RSM) [21], Kriging [22] and radical basis function (RBF) [23] models, etc. The basic mathematical expression of the RBF model is

$$\widetilde{f}(x) = \sum_{i=1}^{N} w_i \varphi(\|x - x_i\|) \tag{24}$$

where $w_i$ is the weight coefficient of the $i$th basis function.

To compare and verify the prediction accuracy of the three models, a cross-validation method was used, where cross-validation means that for the initial sampling data obtained, most of them are used to build a surrogate model, and a small part is used to compare the deviation between the actual and predicted values, and then evaluate the prediction accuracy of the model. There are four commonly used indicators for model accuracy evaluation as follows.

1. Root-mean-square error, *RMSE*

$$RMSE = \sqrt{\frac{1}{m'} \sum_{i=1}^{m'} (y_i - \hat{y}_i)^2} \tag{25}$$

2. $R^2$

$$R^2 = 1 - \frac{\sum\limits_{i=1}^{m'} (y_i - \hat{y}_i)^2}{\sum\limits_{i=1}^{m'} (y_i - \overline{y})^2} \tag{26}$$

where $\overline{y}$ is the mean response value of the $m'$ sample points for validation.

3. Average absolute error, *AAE*

$$AAE = \frac{\frac{1}{m'} \sum\limits_{i=1}^{m'} |y_i - \hat{y}_i|}{\max(y_i) - \min(y_i)} \tag{27}$$

4. Maximum absolute error, *MAE*

$$MAE = \frac{\max(|y_i - \hat{y}_i|)}{\max(y_i) - \min(y_i)} \tag{28}$$

In which, $RMSE$, $R^2$, and $AAE$ are the global prediction accuracy evaluation indexes, and $MAE$ is the local accuracy evaluation index. When $RMSE$ and $AAE$ are closer to 0 and $R^2$ is closer to 1, the global prediction accuracy of the model is higher, and when $MAE$ is closer to 0, the local prediction accuracy of the model is higher.

Based on the sampling point data obtained by OPLHS, the RSM, Kriging and RBF models for $\varepsilon_{v,\max}^{\mathrm{III},p}$ were cross-validated five times, and the results are shown in Table 4.

**Table 4.** Validation results and partial Pareto optimal solutions based on NSGA-II algorithm.

| | **RSM** | | | | **Kriging** | | | | **RBF** | | | |
| --- | --- | --- | --- | --- | --- | --- | --- | --- | --- | --- | --- | --- |
| | *RMSE* | $R^2$ | *AAE* | *MAE* | *RMSE* | $R^2$ | *AAE* | *MAE* | *RMSE* | $R^2$ | *AAE* | *MAE* |
| Test 1 | 0.012 | 0.999 | 0.010 | 0.021 | 0.027 | 0.991 | 0.020 | 0.068 | 0.005 | 1.000 | 0.004 | 0.015 |
| Test 2 | 0.013 | 0.998 | 0.010 | 0.034 | 0.016 | 0.995 | 0.011 | 0.054 | 0.006 | 1.000 | 0.005 | 0.011 |
| Test 3 | 0.006 | 1.000 | 0.005 | 0.013 | 0.037 | 0.987 | 0.024 | 0.083 | 0.004 | 1.000 | 0.003 | 0.008 |
| Test 4 | 0.014 | 0.998 | 0.012 | 0.027 | 0.022 | 0.992 | 0.016 | 0.051 | 0.007 | 0.999 | 0.005 | 0.021 |
| Test 5 | 0.009 | 0.999 | 0.007 | 0.024 | 0.022 | 0.994 | 0.017 | 0.042 | 0.008 | 0.999 | 0.005 | 0.026 |
| Average value | 0.011 | 0.998 | 0.009 | 0.024 | 0.025 | 0.992 | 0.018 | 0.059 | 0.006 | 1.000 | 0.004 | 0.016 |

It was observed that the applicability of different surrogate models was ranked as follows: RBF model > RSM model > Kriging model, and the $R^2$ of the RBF model was always higher than 0.998. It was concluded that the prediction accuracy of the RBF model established based on OPLHS met the requirements, and the subsequent optimization of the SRM grain could be carried out after the surrogate model for $\varepsilon_{v,\max}^{\mathrm{I},p}$ was established by the same method.

## 5. Result of Multiobjective Optimization for SRM Grain

After obtaining the required surrogate model, combined with the computational models of $A_b$ and $\eta_v$, the optimization model shown in Equation (1) is solved based on the NSGA-II algorithm according to the flowchart shown in Figure 14.

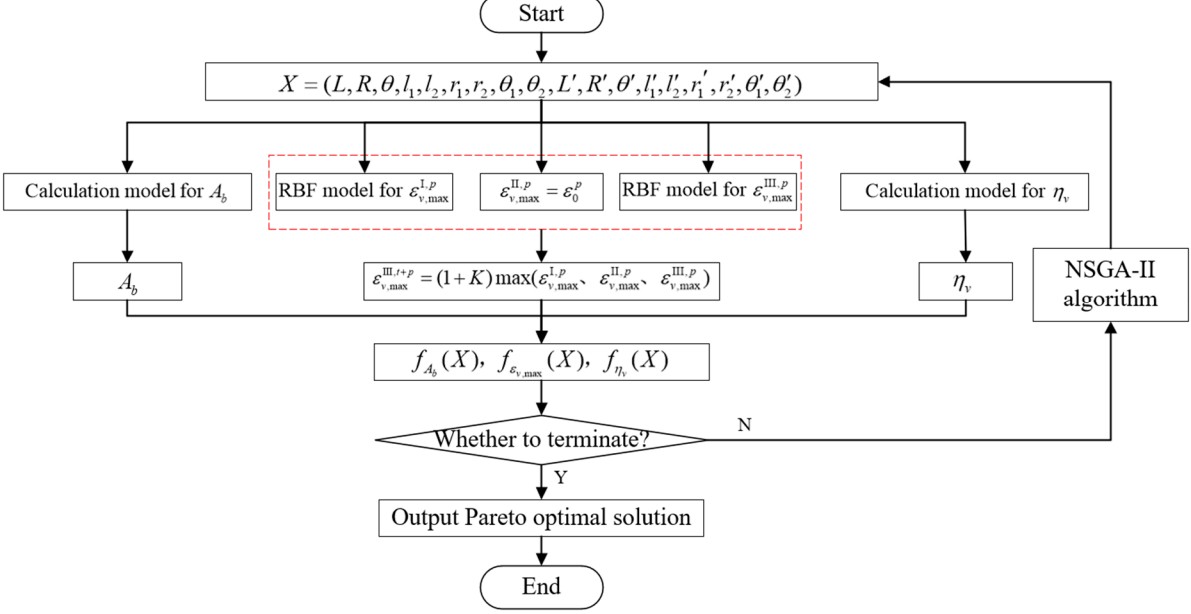

**Figure 14.** Flowchart for solving the multiobjective optimization problem of grain.

NSGA-II is one of the most widely used multiobjective optimization algorithms. The prototype was NSGA, proposed by Srimivas and Deb in 1994; and then K. Deb and S. Agrawal improved NSGA based on NSGA in 2001 and proposed the second-generation of NSGA with elite strategy, namely NSGA-II.

Compared with NSGA, the major difference of NSGA-II is the introduction of crowding degree ranking, which makes the distribution of individuals in each generation of the population more uniform, combined with the Pareto nondominance ranking to determine the fitness of individuals in the population.

Due to the high number of design variables, the number of populations was set to 200 and the number of iterations was set to 50 in order to ensure the diversity of populations. The distribution of the obtained Pareto optimal solution in the space of objective functions was shown in Figure 15.

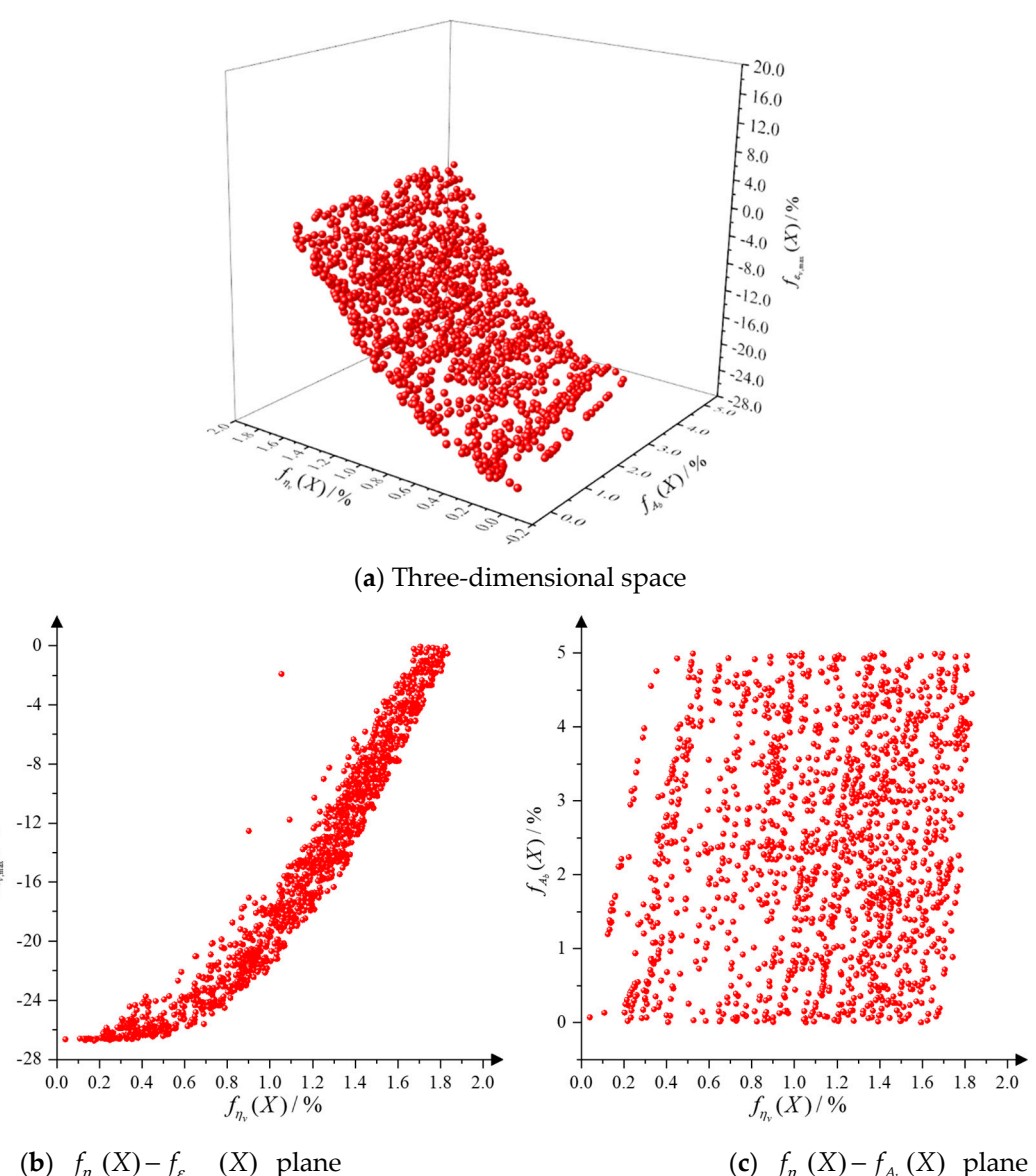

(**a**) Three-dimensional space

(**b**) $f_{\eta_v}(X) - f_{\varepsilon_{v,\max}}(X)$ plane

(**c**) $f_{\eta_v}(X) - f_{A_b}(X)$ plane

**Figure 15.** Schematic diagram of the distribution of Pareto optimal solutions.

From Figure 15, it can be obtained that the optimization method based on the NSGA-II algorithm is able to obtain the complete Pareto front for this problem. In addition, from the distribution of Pareto optimal solutions in the two-dimensional plane of the objective function, it can be seen that there is an obvious constraint relationship between the $\varepsilon_{v,\max}$ and $\eta_v$, and the multiobjective optimization design method proposed in this paper can effectively balance the contradiction.

To verify the accuracy of the multiobjective optimization results of the drug shape based on the surrogate model, the numerical analysis of some of the Pareto optimal solu-

tions was performed, and the comparison results are shown in Table 5, where $\varepsilon_{v,\max}$ is the actual calculated value and $\bar{\varepsilon}_{v,\max}$ is the maximum von Mises strain value of the SRM grain predicted by the surrogate model.

**Table 5.** Partial Pareto optimal solutions and validation results of multiobjective optimization for SRM grain.

| | $f_{\varepsilon_{v,\max}}(X)/\%$ | $f_{\eta_v}(X)/\%$ | $f_{A_b}(X)/\%$ | $\eta_v/\%$ | $\varepsilon_{v,\max}^{t+p}/\%$ | $\bar{\varepsilon}_{v,\max}^{t+p}/\%$ | $\frac{(\varepsilon_{v,\max}-\bar{\varepsilon}_{v,\max})}{\varepsilon_{v,\max}^{t+p}}/\%$ |
|---|---|---|---|---|---|---|---|
| Original | 0.00 | 0.00 | 0.00 | 84.84 | 35.36 | 35.24 | 0.33 |
| | 2.11 | 0.17 | −26.72 | 84.98 | 26.27 | 25.91 | 1.36 |
| | 1.26 | 0.13 | −26.59 | 84.95 | 26.29 | 25.96 | 1.26 |
| | 3.24 | 0.40 | −26.36 | 85.18 | 26.64 | 26.04 | 2.26 |
| Partial | 4.98 | 1.42 | −12.58 | 86.04 | 31.32 | 30.91 | 1.30 |
| Pareto | 0.00 | 1.05 | −1.91 | 85.73 | 33.65 | 34.69 | −3.10 |
| optimal | 0.02 | 0.22 | −26.24 | 85.03 | 26.79 | 26.08 | 2.62 |
| solutions | 0.06 | 0.88 | −17.68 | 85.59 | 29.00 | 29.11 | −0.40 |
| | 0.16 | 0.23 | −25.94 | 85.03 | 26.73 | 26.19 | 2.03 |
| | 0.25 | 1.22 | −13.11 | 85.87 | 31.33 | 30.73 | 1.93 |
| | 0.36 | 0.84 | −20.17 | 85.55 | 28.92 | 28.23 | 2.40 |

The validation results show that the maximum prediction deviation of the surrogate model for $\varepsilon_{v,\max}^{t+p}$ was 3.10% and the average prediction deviation was 1.87%, which confirm the effectiveness and accuracy of the multiobjective optimization method of SRM grain based on the surrogate model.

## 6. Conclusions

In this paper, a multiobjective optimization design of grain is achieved for the SRM with finocyl grain based on parametric modeling and surrogate modeling methods, and the following conclusions are obtained:

(1) The multiobjective optimization of the SRM grain considering the structural integrity, internal ballistic performance, and loading performance was realized, and the complete Pareto optimal solution set was obtained based on NSGA-II with significant optimization effect. Under the constraints, the maximum von Mises strain can be reduced by up to 26.72% and the volume loading fraction can be increased by up to 1.83% compared with that before the optimization, and a series of balanced Pareto optimal solutions were also obtained for the decision maker to choose.

(2) The optimal design method based on parametric modeling and surrogate modeling techniques is significantly superior in efficiency, which can reduce the single calculation time by more than 95% and the number of calculations from 20,000 to 400, with an average prediction deviation of only 1.87%.

(3) Load and geometric parameters were analyzed for the SRM grain under combined temperature and pressure loads, and it was found that the effects of temperature and pressure loads on the von Mises strain at the danger point are superimposable and the ratio between them is fixed; at the same time, the only key geometric parameter affecting the von Mises strain at the danger point is $R, L, \theta$ or $R', L', \theta'$. The resulting conclusions can significantly reduce the work required to build the surrogate model.

**Author Contributions:** Conceptualization, Z.S. and Q.M.; methodology, Z.S.; software, Q.M.; validation, Q.M.; formal analysis, Q.M.; investigation, H.Z.; resources, Z.S.; data curation, H.S.; writing—original draft preparation, Q.M.; writing—review and editing, H.Z.; visualization, H.S.; supervision, Z.S.; project administration, H.S.; funding acquisition, Z.S. All authors have read and agreed to the published version of the manuscript.

**Funding:** This research was funded by the National Natural Science Foundation of China (Nos. 11872372) and Hunan Provincial Natural Science Foundation of China (Nos. 2021JJ10046).

**Data Availability Statement:** Not applicable.

**Conflicts of Interest:** The authors declare no conflict of interest.

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
