# Peer review of "Multiobjective Optimization Method of Solid Rocket Motor Finocyl Grain Based on Surrogate Model"

_aerospace, doi:10.3390/aerospace9110679_

Round 1

Reviewer 1 Report

The manuscript presents a new parametrized model method for structural analysis of high-loading solid rocket motors. In this method, some parameters are kept constant, and some hypotheses are considered. Based on the results, seems that there is low accuracy loss and significant decrease of the computational costs.

The authors state that long-term experience indicates that temperature and pressure loads have the most significant impact on the structural integrity of the SRM, so gravity and vibration loads are not considered. These loads are significant in large scale motors, so wouldn't be interesting to validate the script with other (experimental) rockets?

On page 6, line 174, it is mentioned the 'drug form' optimization, but this term is not clear. Could you please clarify?

What do the authors mean by 'manual modeling time is about tens of hours'. Is the manual modeling mentioned the process of drawing and creating a mesh for the system? If so, with the proper software, this effort hardly takes 'tens of hours'.

In the Function of load setting, why were these specific parameters (regarding temperature, time and pressure) chosen?

Figs. 8 and 9 lack the units for temperature and pressure, respectively.

Sentence on lines 315 and 316 should be rewritten as it is confusing.

There is a comma left alone at the beginning of line 327.

Remove the dot after 'Table 5' in line 415.

Author Response

The authors wish to thank the editors and reviewers for their time in effort in reviewing our manuscript. We hope the changes listed have made the manuscript suitable for publication and we look forward to your response, and the specific responses to reviewers' comments are shown in the document.

Reviewer 2 Report

See attachment.

Author Response

The authors wish to thank the editors and reviewers for their time in effort in reviewing our manuscript.We hope the changes listed have made the manuscript suitable for publication and we look forward to your response, and the specific responses to reviewers' comments are shown in the document.
